# Non-conventional mechanism of ferroelectric fatigue via cation migration

Anton V. Ievlev [1], Santosh KC [2], Rama K. Vasudevan[1], Yunseok Kim[3], Xiaoli Lu [4], Marin Alexe[5], Valentino R. Cooper [2], Sergei V. Kalinin [1] & Olga S. Ovchinnikova [1]

The unique properties of ferroelectric materials enable a plethora of applications, which are hindered by the phenomenon known as ferroelectric fatigue that leads to the degradation of ferroelectric properties with polarization cycling. Multiple microscopic models explaining fatigue have been suggested; however, the chemical origins remain poorly understood. Here, we utilize multimodal chemical imaging that combines atomic force microscopy with time-of-flight secondary mass spectrometry to explore the chemical phenomena associated with fatigue in $PbZr_{0.2}Ti_{0.8}O_3$ (PZT) thin films. Investigations reveal that the degradation of ferroelectric properties is correlated with a local chemical change and migration of electrode ions into the PZT structure. Density functional theory simulations support the experimental results and demonstrate stable doping of the thin surface PZT layer with copper ions, leading to a decrease in the spontaneous polarization. Overall, the performed research allows for the observation and understanding of the chemical phenomena associated with polarization cycling and their effects on ferroelectric functionality.

[1] The Center for Nanophase Materials Sciences, Oak Ridge National Laboratory, Oak Ridge, TN 37831, USA. [2] Materials Science and Technology Division, Oak Ridge National Laboratory, Oak Ridge, TN 37831, USA. [3] School of Advanced Materials Science and Engineering, Sungkyunkwan University, Suwon 16419, Republic of Korea. [4] The State Key Discipline Laboratory of Wide Band Gap Semiconductor Technology, Xidian University, 710071 Xi'an, China. [5] Department of Physics, University of Warwick, Coventry CV4 7AL, UK. Correspondence and requests for materials should be addressed to A.V.I. (email: ievlevav@ornl.gov) or to O.S.O. (email: ovchinnikovo@ornl.gov)

For the last several decades ferroelectric materials have attracted much interest due to the rich spectrum of properties that enable a wide range of practical applications from mechanical actuation to data storage and light conversion[1–9]. However, many of these applications are hindered by ferroelectric fatigue, which leads to a reduction of the switchable polarization with increasing number of polarization reversal cycles[10–12]. To date, the origin of the fatigue is not fully understood. The degradation has been found to be dependent upon multiple parameters including doping, material of electrode, defect structure, and temperature[13–16]. Multiple microscopic models have been suggested which attribute it to the generation of crystallographic defects[12,13,17–19], the redistribution of existing defects[20–22], the reduction of effective external electric fields[23,24], or a modification of the switching process[25–27].

However, the chemical aspects of fatigue are still poorly understood with studies mostly limited to electron microscopy research, which do not allow for examination of the connection between material functional response and local chemical signatures in-situ[19,28–30]. Multimodal chemical imaging offers an alternative for investigating the link between material functionality and local chemistry. This approach combines different complimentary imaging techniques, like atomic force microscopy (AFM), in order to provide information about functional material responses and mass spectrometry chemical imaging to allow for the precise mapping of the chemical composition of the surface and in the bulk. In particular, Time-of-Flight Secondary Ion Mass Spectrometry (ToF-SIMS) is well suited for this type of chemical analysis. ToF-SIMS utilizes a primary focused ion beam to release and ionize material, which can then be detected with a time-of-flight analyzer to identify the mass-to-charge ratio[31,32]. Combined with additional sputtering guns ToF-SIMS allows for chemical studies of surfaces and in the bulk with sub-100-nm lateral and sub-nanometer depth resolution. Recent studies have demonstrated ToF-SIMS's applicability for chemical investigations in ferroelectric materials[33–39].

In this work, we utilize multimodal chemical imaging that combines AFM with ToF-SIMS to study the chemical phenomena associated with fatigue in ferroelectric thin films. Experiments are carried out using $PbZr_{0.2}Ti_{0.8}O_3$ thin films with an array of sub-micrometer Au/Cu top electrodes and a $SrRuO_3$ bottom electrode to form an array of nanocapacitors. The AFM tip inside the ToF-SIMS vacuum chamber is used to locally bias individual capacitors to switch and cycle them. The properties of the cycled capacitors are further investigated using local hysteresis loop measurements and ToF-SIMS 3D chemical mapping. AFM studies reveal a decrease in the coercive bias along with a reduction of the total piezoresponse when the number of switching cycles exceeded ten thousand. Subsequent 3D chemical imaging indicates that this behavior is correlated with the quasi-doping of the film due to the introduction of copper ions into a few nanometer-thick layer of PZT. These results are consistent with DFT calculations showing a corresponding decrease in the spontaneous polarization with increasing Cu doping concentration. The explored chemical phenomena are important for understanding the fundamental properties of ferroelectric materials and their practical applications.

## Results

**Polarization switching and cycling.** In this work we used a 90-nm-thick ferroelectric film of lead zirconate titanate $PbZr_{0.2}Ti_{0.8}O_3$ (PZT) grown on a $SrTiO_3$ substrate with a $SrRuO_3$ conductive buffer layer. Arrays of Au/Cu sub-micron sized circular electrodes (average radius $r_{cap} = 370$ nm and area $S_{cap} = 0.43$ μm$^2$) were deposited on the surface of the film (Fig. 1a). The

thickness of the electrodes was 25 nm, comprised of 10 nm of a top gold layer and 15 nm of a copper interfacial layer. Details of the sample preparation can be found in ref. [40]. Copper is considered as a cheaper alternative to noble metals for electrodes in ferroelectric based devices[41]. Altogether, the bottom $SrRuO_3$ and top electrodes in this configuration formed an array of nanocapacitors, that could be ferroelectrically switched by the application of an external electric field.

Local polarization switching was accomplished by applying a bias to a conductive AFM tip in contact with the top electrode of an individual nanocapacitor. This measurement was combined with piezoresponse force microscopy (PFM), allowing for correlated nanoscale imaging of the resultant ferroelectric domain structure[42–44]. Therefore, the AFM tip was used for both switching and controlling the spontaneous polarization in individual nanocapacitors. Using this approach, we demonstrated the ability to switch the polarization inside individual capacitors through the application of ±4 V (Fig. 1b–g). These experiments showed inversion of the PFM phase over the whole capacitor (Fig. 1e–g) but did not indicate any changes in capacitor topography (Fig. 1b–d). The switched capacitor is labeled by a red arrow in Fig. 1e–g. The ability to switch a larger number of capacitors during scanning with an applied AFM tip bias has also been demonstrated (Supplementary fig. 1).

Initial ToF-SIMS investigations of pristine capacitor arrays showed the expected distribution of base chemical elements, with $Au^+$ and $Cu^+$ present in the top electrode only, $Pb^+$, $Zr^+$, and $Ti^+$ inside the film, and $Sr^+$ and $Ru^+$ in the buffer layer (Supplementary fig. 2). $Cu^+$ was found along all regions of the electrode including the top $Au^+$ layer; suggesting significant $Cu^+$ ion intermixture throughout the electrode. The lateral resolution of the ToF-SIMS chemical imaging was ~120 nm, which was insufficient for mapping changes in the chemistry of individual electrodes.

To study ferroelectric fatigue in the PZT thin-film, we cycled the capacitors by the application of bipolar triangular pulses with an amplitude of 8 V. This roughly corresponds to the amplitude of an applied electric field of 89 kV/mm. The evolution of the ferroelectric response in the capacitor was measured using PFM local hysteresis loops[43,44] after a certain number of applied switching cycles. In these measurements, PFM signals were measured during the application of the switching field, which allowed for the control of the spontaneous polarization in the region underneath the AFM tip. Combined piezoresponse signal $PR = A \cdot \cos(\theta)$, was used for analysis of the switching behavior, where $A$ is the PFM amplitude and $\theta$ is the PFM phase. The experiments showed significant changes in the shape of the hysteresis loop with successive switching cycles (Fig. 2a). To quantitively characterize these changes, we calculated the values of the forward and reverse coercive voltage ($V^+$ and $V^-$, respectively) and the forward and reverse saturation responses ($R^+$ and $R^-$, respectively), as defined in ref. [45]. Using these extracted values, we calculated the values of the coercive field $E_c = (V^+ - V^-)/2t$, imprint field $E_{im} = (V^+ + V^-)/2t$ and maximal switchable response $R_m = R^+ - R^-$, where $t = 90$ nm is the film thickness (Fig. 2b–d).

An analysis of the extracted parameters revealed two main trends. First, the coercive field increased from 20 to 35 kV/mm after 5000 pulses, with no significant changes in $E_{im}$ and $R_m$. Further cycling revealed significant changes in all parameters. In particular, the coercive field decreased to 15 kV/mm (Fig. 2b), while the imprint field increased from ~0 to ~9 kV/mm (Fig. 2c) and the maximal switchable response dropped by more than a factor of 4 (Fig. 2d). Such observations are attributes of ferroelectric fatigue and the degradation of ferroelectric properties in the cycled capacitor. To exclude the influence of the degradation of the AFM tip, we performed measurements on a

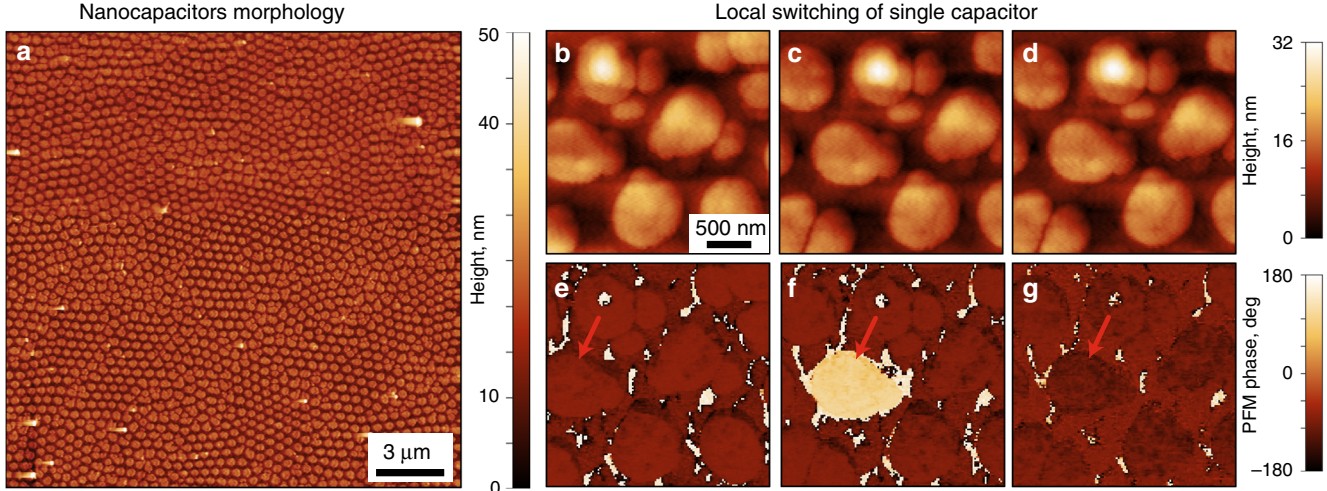

**Fig. 1** AFM studies of PZT nanocapacitors. **a** Topography of the pristine sample. **b–g** Polarization switching of a single capacitor via an AFM tip: **b**, **e** before switching; **c**, **f** switching with +4 V; **d**, **g** back switching with −4 V. Signals of **b–d** topography and **e–g** PFM phases

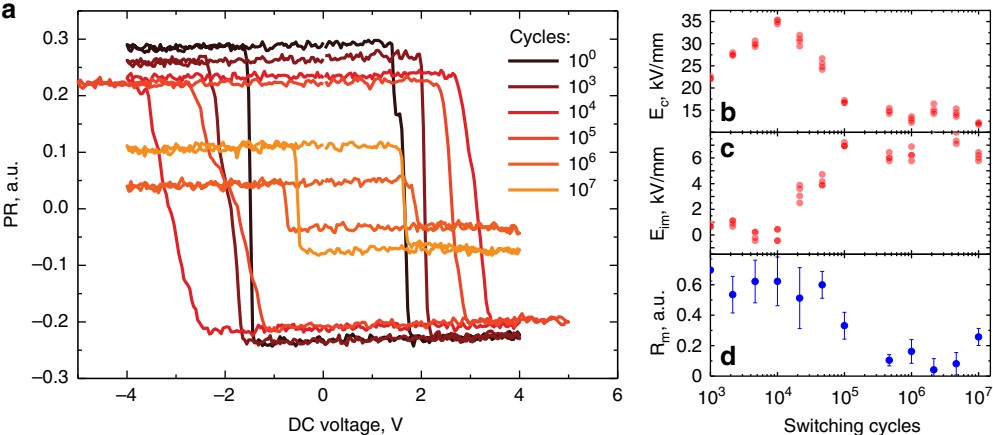

**Fig. 2** Hysteresis loops of the single capacitor measured via AFM. **a** Evolution of the hysteresis loop with number of switching pulses; **b** coercive field $E_c$, **c** imprint field $E_{im}$ and **d** maximal switchable response $R_m$ (error bars calculated as standard deviation of 50 points dataset) as a function of number of switching pulses

fresh capacitor using the same tip. The resulting hysteresis loops demonstrated piezoresponse and $R_m$ similar to the non-fatigued loops of the first capacitor (Supplementary fig. 3), confirming that the observed hysteresis loop evolution is related to the degradation of the film properties, and not tip degradation.

**Chemical studies of ferroelectric fatigue**. To gain insight into the chemical phenomena associated with the observed ferroelectric fatigue, ToF-SIMS chemical mapping was further employed. Due to the lack of spatial resolution and weak SIMS signals, we were unable to study the chemistry associated with the ferroelectric fatigue of an individual capacitor. To resolve this issue, we performed cycling of a larger area of capacitors. In these experiments, square areas of $5 \times 5 \, \mu m^2$ were scanned in AFM contact mode with an applied AC tip bias. The amplitude of the bias was 8 V, with a frequency $f_{ac} = 1–20$ kHz. The total scanning time $T_{scan}$ was tuned to maintain an average number of the switching pulses $N$ applied to every capacitor within the scanned area:

$$T_{scan} = \frac{S_{scan}}{S_{cap}} \cdot \frac{N}{f_{ac}} \quad (1)$$

where $S_{scan} = 25 \, \mu m^2$ is the total scan area and $S_{cap} = 0.43 \, \mu m^2$ is the average area of a single capacitor.

In total, six areas were prepared in this fashion. The first, was switched by DC voltage only, while the remaining five were cycled with a total number of switching pulses per capacitor ranging from $10^2$ to $10^6$. Panoramic PFM images of the resulting structures are presented in Fig. 3. Detailed images of the switched and cycled areas can be found in Supplementary fig. 4.

Similar to the local hysteresis measurements, PFM imaging revealed changes in the resulting domain structure. For up to $10^4$ cycles, the PFM amplitude signal (Fig. 3a) did not show any changes associated with the cycling, while the PFM phase (Fig. 3a) had a random distribution associated with the last polarity of the bias applied to a particular capacitor. However, increasing the number of cycles led to the appearance of dark regions in the PFM amplitude signal, associated with a decrease in the piezoresponse of the cycled capacitors (Supplementary fig. 5). This result is in good agreement with the hysteresis measurements, which showed a drop in the total switchable response $R_m$ above $10^4$ cycles (Fig. 2d).

The chemical changes of the cycled regions were further studied using ToF-SIMS 3D chemical imaging with a

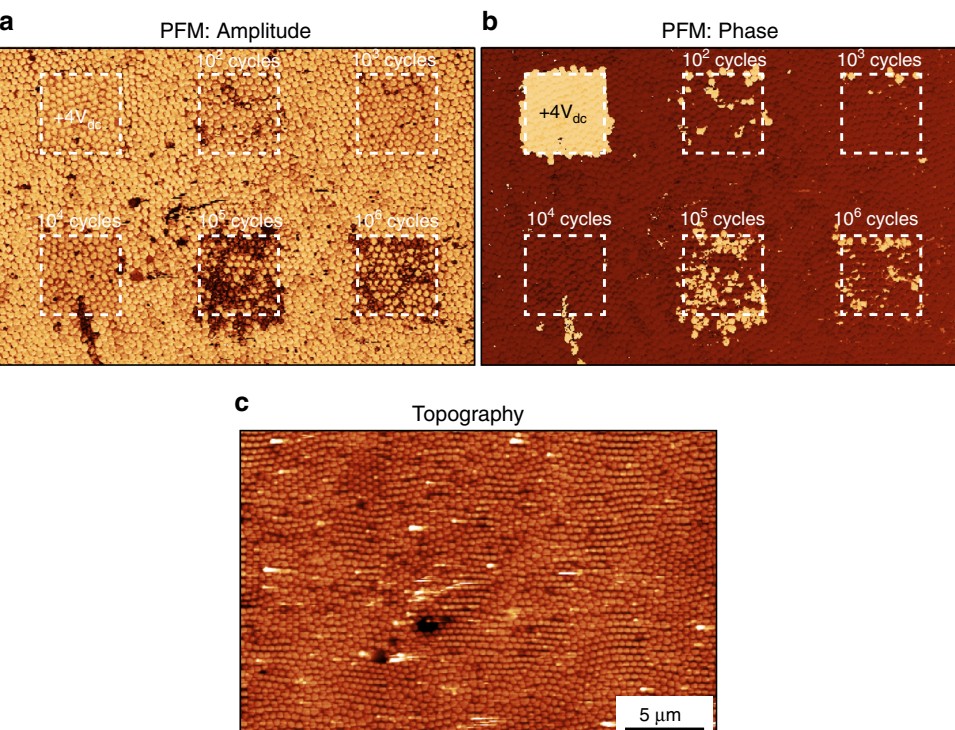

**Fig. 3** PFM imaging of regions cycled with different numbers of switching pulses (from $10^2$ to $10^6$ as labeled). **a** PFM amplitude; **b** PFM phase; and **c** topography

$Bi_3^+$ primary source and a $O_2^-$ sputter source. Chemical imaging was carried out using the high lateral resolution imaging mode, which provides a spatial resolution of about 120 nm and a mass resolution of $\Delta m/m < 300$. A low value of the mass resolution can hamper the identification of elements and clusters with similar masses. In particular, in the studied sample, the peak of $Cu^+$ with a $m/z = 62.929$ u overlaps with the peak of $^{47}TiO^+$, with a $m/z = 62.946$ $u$ (Supplementary fig. 6). To resolve this issue, we used the intensity of the $TiO^+$ ($^{48}TiO^+$) peak, which did not overlap with other $Cu^+$ isotopes, and values of $TiO^+$ isotopic abundances (73.56% for $TiO^+$ and 7.44% for $^{47}TiO^+$) to correct for the $Cu^+$ intensity as:

$$I_{Cu}^{corr} = I_{63} - \frac{7.44}{73.56} I_{TiO} \qquad (2)$$

where, $I_{Cu}^{corr}$ is the corrected intensity of $Cu^+$; $I_{63}$ is the measured intensity of the peak at 63 u, combining $Cu^+$ and $^{47}TiO^+$; $I_{TiO}$ is the intensity of the $TiO^+$ peak.

Here and below all chemical maps of $Cu^+$ spatial distributions are plotted using the corrected value of intensity $I_{Cu}^{corr}$. An analysis of the resultant chemical data allowed for the identification of all expected elements of the electrode, PZT film and buffer $SrRuO_3$ layer. No chemical changes were detected within the regions switched by DC voltage, and cycled with $10^2$ and $10^3$ switching pulses (Fig. 4a, Supplementary fig. 7). Changes were revealed only starting at $10^4$ cycles. In particular, we found significant penetration of the $Cu^+$ ions from the top copper electrode into the PZT thin-film (Fig. 4b). The penetration depth increased with the number of cycles. This was also accompanied by a slight decrease in the $Pb^+$ concentration inside the cycled regions (Fig. 4c). The spatial distribution of other chemical elements of PZT and the buffer layer were not modified by the cycling process.

To measure the depth of $Cu^+$ penetration into the PZT film we plotted depth profiles of pristine samples and cycled regions with different numbers of switching pulses (Fig. 4d). Depth calibration has been performed using the known thickness of the PZT film (90 nm), and with the assumption of homogeneity of the sputtering rate within the film and electrode. The depth profiles clearly show the propagation of the $Cu^+$ ions inside the film (Fig. 4d). To measure the propagation depth, we further subtracted the profile of the pristine area from the profiles of the cycled areas (Fig. 4e). The penetration depth was defined as a point where the differential profiles dropped below the noise level. The resulting penetration depth ranged from 8 nm for $10^4$ cycles to about 16 nm for $10^6$ cycles (Fig. 4e inset). No significant changes were found in the depth profile averaged over the area cycled by $10^3$ switching pulses.

To understand the role of the oxygen vacancies in the fatigue phenomenon we studied the chemical maps of the $TiO^+$ spatial distribution (Supplementary fig. 8b, e). To uncouple oxygen behavior from titanium we calculated the ratio of $TiO^+/Ti^+$ (Supplementary fig. 8c, f). An analysis of the resulting maps did not reveal any spatial correlations within the fatigued regions.

**Numerical simulations of PZT doping by Cu.** To gain further insight into the chemical phenomena associated with Cu migration into the PZT structure, we performed density functional theory (DFT) calculations of PZT doped by $Cu^+$ ions. In these simulations, we used supercells of $PbZr_{0.125}Ti_{0.875}O_3$ to examine the substitutional doping of Pb with Cu (Fig. 5a). Doping concentrations ranged from 12.5 to 50% ($Pb_{0.875}Cu_{0.125}Zr_{0.125}Ti_{0.875}O_3$ to $Pb_{0.5}Cu_{0.5}Zr_{0.125}Ti_{0.875}O_3$). Figure 5b depicts the band gap as a function of Cu doping, for the doped PZT system. Here we find that for doping concentrations ≥37.5% the band gap closes and the material becomes metallic. We note that in all cases the substitution of Cu on the Pb site is energetically favorable

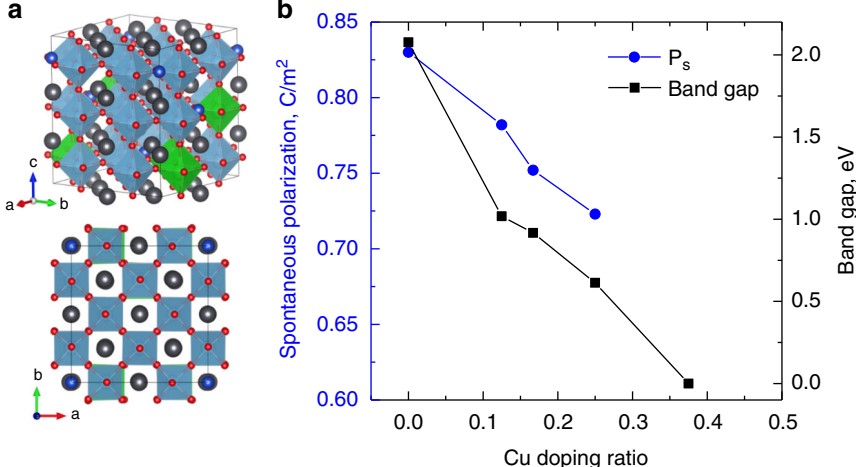

**Fig. 4** ToF-SIMS investigation of the chemical phenomena associated with fatigue in a PZT. **a** XY map of $Cu^+$ distribution averaged over first 5 nm depth layer of PZT right underneath the electrode; **b**, **c** XZ cross-sections of **b** $Cu^+$ and **c** $Pb^+$ averaged over the area of regions cycled with $10^4$ to $10^6$ pulses. **d** Depth profiles of $Cu^+$ distribution averaged over the pristine and cycled regions. **e** Differential profile of the $Cu^+$ concentration inside the cycled regions with respect to the pristine film; (inset) $Cu^+$ penetration depth vs number of applied switching pulses

**Fig. 5** DFT simulations of Cu-substituted $PbZr_{0.125}Ti_{0.875}O_3$. **a** Relaxed supercell of $Pb_{1-x}Cu_xZr_{0.125}Ti_{0.875}O_3$ with $x = 0.125$ (black balls—Pb, blue balls—Cu, red balls—O, blue octahedra—Ti, green octahedra—Zr) and **b** calculated spontaneous polarization (blue circles) and electronic band gap (black squares) as a function of Cu doping concentration ($x$)

with formation energy differences of ~2.1 eV per Cu ion. In all cases, we computed large off-centering of $A$- and $B$-site cations within their octahedral cages (see Supporting Information for the average off-centering values) suggesting large polarizations should persist within the unit cell. However, it is only possible to compute the polarization in the insulating case. Here we find that the polarization decreases significantly with increasing Cu content; going from 0.83 C/m$^2$ (at $x = 0$) to 0.72 C/m$^2$ (at $x = 25\%$) (Fig. 5b). The decrease in polarization and electronic band gap are in good agreement with the experimental results that indicate a loss of the ferroelectric behavior as a function of cycling. The full table of the results from the calculations can be found in Supplemental Materials (Supplementary table 1).

## Discussion

Summarizing, both ToF-SIMS and PFM local hysteresis measurements showed significant changes in PZT capacitor properties after 10$^4$ switching cycles. At the same time, an initial increase in the coercive field observed up to 5000 cycles did not show any chemical changes within the film. This trend can be attributed to the known wake-up effect, leading to an increase of the switching charge during the first few thousand switching cycles[46,47]. The unchanged chemistry provides evidence for recent interpretations of the origins of ferroelectric fatigue that indicate significant effects due to the depinning process or the reduction of locked localized charges within the ferroelectric medium[47]. A further increase in the number of switching cycles led to fatigue, signaled by a significant decrease in the maximal switchable response and coercive field as well as the appearance of an imprint field. Chemical studies revealed a corresponding migration of Cu$^+$ ions from the electrode into the film, accompanied by a slight depletion of Pb$^+$. This result can be attributed to effectively doping the PZT surface layer by Cu$^+$ ions. DFT simulations confirmed the stability of the resulting doped structure and showed a corresponding decrease in spontaneous polarization. Furthermore, DFT simulations revealed a transition to a metallic state above 25% of Cu concentration. This potentially explains the decrease in the coercive fields for large numbers of the switching cycles due to an enhancement of the surface layer conductivity and corresponding increase in the effective electric field.

On the other hand, the 4-fold drop in switchable piezoresponse cannot be addressed due to the spontaneous polarization decrease in the top 16 nm (17% of the film thickness) layer of PZT, as the expected changes would have to be of the same order. One explanation could be that the doping is inhomogeneous, and, thereby coupled with the formation of small regions with lower spontaneous polarizations, which can play the role of pinning centers. Those regions would decelerate the motion of domain walls and lead to the formation of frozen domain structures, which in turn would decrease the switchable charge[26,48]. Alternatively, doped regions may induce the formation of defect states, with the same stoichiometry, which are undetectable with ToF-SIMS, but can significantly alter the switchable polarization. At the same time, local oxygen concentration changes were not found within the ToF-SIMS detection limits. This implies that the concentration of oxygen vacancies, which are believed to be the main driving force of the fatigue, do not change significantly. However, local ordering of oxygen vacancies, undetectable by ToF-SIMS, needs further investigations.

In conclusion, in this work we investigated the chemical phenomena associated with ferroelectric fatigue in lead zirconate titanate nanocapacitors using multimodal chemical imaging. Using an AFM tip, we were able to locally cycle and switch the spontaneous polarization of individual nanocapacitors. In particular, using this approach we observed significant changes in the coercive field, maximal switchable response and imprint field when the number of switching cycles exceeding 10$^4$. Subsequent ToF-SIMS measurements revealed a corresponding migration of copper ions from the top electrode into the film structure, which was accompanied by the depletion of lead ions from the film. At the same time changes in oxygen concentration were not observed. Altogether these results revealed quasi-doping of the top PZT layer by the copper ions from the electrode. DFT simulations confirmed the stability of the resultant structure; indicating a corresponding reduction in the spontaneous polarization. This allowed us to attribute the observed ferroelectric fatigue to the formation of pinning center or defect structures along the doped regions, which in turn decrease the switchable polarization. The explored phenomena sheds light on the chemical aspects of ferroelectric functionality and are important for the development of novel ferroelectric devices.

## Methods

**Sample preparation.** A 90-nm thick epitaxial Pb(Zr$_{0.2}$Ti$_{0.8}$)O$_3$ (PZT) thin-film was prepared by pulsed laser deposition on SrRuO$_3$ (SRO) (001)/SrTiO$_3$ (STO) substrate. A self-ordered ultrathin anodic aluminum oxide (AAO) mask was prepared by anodization of aluminum and placed on the top of the PZT thin-film. Then, 25-nm thick Au/Cu (Au: 10 nm and Cu: 15 nm) was subsequently deposited by thermal evaporation. Finally, the nanoelectrodes with a diameter of around 370 nm were obtained by removing the AAO mask. The detailed information of the PZT thin films and AAO masks can be found elsewhere[40,49].

**Atomic force microscopy.** Atomic Force Microscopy measurements was carried out using Nanoscan AFM instrument inside the ToF-SIMS vacuum chamber. Scanning was performed using Nanosensor PPP-EFM tips with a platinum-iridium coating. PFM imaging was realized using AC voltage with amplitude 250 mV and frequency 300 kHz applied to the tip and utilizing a SR865A lock-in amplifier (Stanford Research). For switching experiments DC or AC voltage was applied to the top sub-micrometer sized electrode via tip brought into the contact. Amplitude of AC voltage was 8 V, frequency ranged from 1 to 20 kHz. All measurements were done in vacuum 8–9 × 10$^{-9}$ mbar.

**Time-of-flight secondary ion mass spectrometry.** ToF-SIMS measurements were done using a TOF.SIMS.5-NSC instrument, using a Bi$_3^+$ ion gun (30 keV energy, 0.49 nA current) as the primary ion source and an O$_2^-$ ion gun (1 keV energy, 120 nA current, 20 μm spot size) as the sputter source. For the imaging ToF-SIMS experiments we used a mode with high spatial resolution—Bi spot size ~120 nm, energy 30 keV, current 0.49 nA, and mass resolution $\Delta m/m = 100–300$. ToF-SIMS measurements were performed in the non-interlaced mode, where every scan of chemical analysis with bismuth primary source was followed by sputtering using O$_2^-$ ion gun. A low energy electron flood gun was used for charge compensation between cycles. The vacuum level in the ToF-SIMS during the measurements ranged from 5 to 9 × 10$^{-9}$ mbar.

**Ab-initio DFT simulations.** Density functional theory (DFT) calculations were performed using the Vienna *Ab-initio* Simulation Package (VASP) with the local density approximation (LDA) and projector augmented-plane-wave (PAW) potentials with electronic configurations of Cu:3p$^6$4d$^9$4s$^2$, Pb:5d$^{10}$6s$^2$6p$^2$, Zr:4s$^2$4p$^2$4d$^2$5s$^2$, Ti:3s$^2$3p$^2$3d$^2$4s$^2$, and O:2s$^2$2p$^4$ [50]. A cutoff energy of 500 eV and a $2\sqrt{2} \times 2\sqrt{2} \times 3$ (120 atom) supercell model of Pb$_{1-x}$Cu$_x$Zr$_{0.125}$Ti$_{0.875}$O$_3$ was studied with $x = 0$, 3/24, 4/24, 6/24, 9/24, and 12/24. To construct our supercell model we employed the special quasirandom structure (SQS) approach to define a random arrangement of cations[51]. Previously, we demonstrated that this approach was able to properly capture the atomic distortions present in a disordered, perovskite solid solution[52]. A Monkhorst Pack $k$-point grid of $2 \times 2 \times 2$ was used for the Brillouin zone integration during structural optimization. The structural optimizations were achieved by allowing the atoms and lattice to relax until all the forces on each atomic site were below 5 meV Å$^{-1}$ and simultaneously achieving a total energy convergence of 10$^{-6}$ eV. (See Supporting Information for final structures). Born effective charges ($Z^*$) were computed using density functional perturbation theory (DFPT) as implemented in the VASP code and are listed in the Supporting Information Table. S1. The ferroelectric polarization for Cu doping concentrations between 0 and 25% were estimated using the Born effective charges and off-center displacements.

## Data availability

The data that support the findings of this study are available from the corresponding author upon reasonable request.

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

## Acknowledgements

This research was supported by the Laboratory Directed Research and Development Program of Oak Ridge National Laboratory, managed by UT-Battelle, LLC, for the U. S. Department of Energy (A.V.I., O.S.O.). A portion of this research, including computational aspects (S.K.C., V.R.C.) was supported by the U.S. Department of Energy, Office of Science, Basic Energy Sciences, Materials Sciences and Engineering Division (R.K.V., S.V. K.). All calculations were performed at the National Energy Research Scientific Computing Center, a DOE Office of Science User Facility supported by the Office of Science of the U.S. Department of Energy under Contract No. DE-AC02-05CH11231. This research was conducted at the Center for Nanophase Materials Sciences, which is a DOE Office of Science User Facility, and using instrumentation within ORNL's Materials Characterization Core provided by UT-Battelle, LLC under Contract No. DE-AC05-00OR22725 with the U.S. Department of Energy.

## Author contributions

A.V.I. has obtained and analyzed the experimental AFM and ToF-SIMS data. Y.K., X.L., R.K.V., and M.A. prepared sample of PZT film with nanoelectrodes. S.K.C. and V.R.C. performed numerical DFT simulations. S.V.K. and O.S.O. directed research. All authors wrote the manuscript.

## Additional information

**Competing interests:** The authors declare no competing interests.

