## [Peer Review File · Nature Communications]

Reviewers' comments:

Reviewer #1 (Remarks to the Author):

The manuscript contains new, relevant insights that are worth publishing in Nature Communications. As the title points out, the authors study the non-conventional mechanism of ferroelectric fatigue via cation migration. Although the title is covering a wide field, the study is focused on the PZT thin film only. First, the atomic force microscopy (AFM) is used to induce and measure the fatigue behaviors from one or multiple nanocapacitors. Then, the time-of-flight secondary ion mass spectrometry (ToF-SIMS) is utilized to record the cation migration during the fatigue process. The conclusion was made that the degradation of ferroelectric properties was correlated with a local chemical change associated with the penetration of Cu^+ ions from the electrode into the PZT structure, while the impact from oxygen vacancies was found to be insignificant. The work is well organized, and the conclusion is challenging the state-of-art points of view on the ferroelectric fatigue. Although both AFM part and ToF-SIMS part are designed well, the logic behind the correlation of these two parts can be improved.

In line 242-246

“As such, one can conclude, that fatigue in the studied system is mostly caused by effective doping of the PZT film due the migration of copper ions from the electrode.”

The insignificance of one driving force, which is oxygen vacancies contribution in this work, will not directly make another driving force, which is cation migration, more significant.

This also raises more questions:

1. Is Cu ion special in the ferroelectric fatigue processing? How about other cations used for electrode?
2. The DFT simulation in this work clearly shows that spontaneous polarization is significantly reduced with increasing Cu in $\text{Pb}(1-X)\text{Cu}_x\text{TiO}_3$. However, Is it reasonable to compare the Cu ion concentration in PZT thin film with that in $\text{Pb}(1-X)\text{Cu}_x\text{TiO}_3$?
3. Suppose that Cu ion really is the origin of the fatigue, how does the limited Cu ion affect the whole PZT film or single nanocapacitor? According to the Fig.2b and line 226, the Cu ion penetration depth ranged from 8 nm to 16 nm and the PZT thin film is about 90 nm thick.

Reviewer #2 (Remarks to the Author):

Based on the combination of PFM and ToF-SIMS, the authors investigated polarization fatigue phenomenon of PZT capacitors in nanoscale. The authors observed the migration of Cu⁺ ions into ferroelectric. The results are interesting. However, I don't think the authors well demonstrated the correlation between Cu⁺ ion migrations with polarization fatigue. The current version is not suitable for publication in nat. Commun. Some of my comments are listed below.

1) though the authors observed the migration of Cu⁺ ions in fatigued PZT nanocapacitors, there are no further evidences to prove that this migration results in ferroelectric fatigue. That is, this migration may be only one of experimental phenomena observed during fatigue process, but is not the reason of ferroelectric fatigue. Though the DFT simulation indicated that Cu doping in PbTiO₃ resulted in the decrease of polarization, I am not very sure whether the doping in simulation is the same or similar as the migration in experiment.

2) In fact, the migrations of metal ions into insulating layer in MIM systems are well observed and studied, which is also regarded as one possible mechanism forming resistive devices (sometimes called conductive filaments). From this viewpoint, Cu⁺ ion migration may result in the increase of conductivity of the PZT thin layer near the Cu electrode. This may result in the decrease of effective PZT thickness and thus the decrease of apparent coercive field.

3) about Figure 1, the authors should indicate which nanocapacitor was polarized in text.

4) about Figure 2a and relevant text, here 'PFM signals' means PFM phase signals? The hysteresis loops are usually obtained from PFM phase signals. Why the authors used the term 'piezoresponse' instead of 'PFM phase'? Why the change in PFM phase signals can respond the max. switchable response R_m ?

5) p7, line 129, 8V voltage is applied to 90nm thick PZT layer, how the authors get the field of 114 kV/mm?

6) Figure S3b, the hysteresis loops from samples 1 and 2 show very different coercive fields and a little different piezoresponse. Why the authors explained that 'The resulting hysteresis loops demonstrated behavior similar to the non-fatigued loops of the first capacitor (Supplemental Materials, Fig. S3)' in the main text (Line 154, p9)?

7) about the equation in p9, Line 165. Since not all local area is covered by electrodes, so Scan/Scap is not equal to the number of nanocapictors. Thus the calculated T_{scan} is not correct.

8) Figure 3a, p10, Line 182-183, 'However, increasing the number of cycles lead to the appearance of dark regions in the PFM amplitude signal, associated with a decrease in the piezoresponse of the cycled capacitors.' From Figure 3a, the local area after 10^6 cycles seems to have even more electro-active nanocapacitors than the local area after 10^4 cycles. Why?

8) in the whole text, what is the meaning of ‘...averaged over 5 nm surface layer of PZT’?

9) Figure 5a, atom species should be labeled.

10) in method part, the authors need to give more details on how to use AAO templates to form these nanoelectrodes.

Detailed response

Reviewer #1 (Remarks to the Author):

The manuscript contains new, relevant insights that are worth publishing in Nature Communications. As the title points out, the authors study the non-conventional mechanism of ferroelectric fatigue via cation migration. Although the title is covering a wide field, the study is focused on the PZT thin film only. First, the atomic force microscopy (AFM) is used to induce and measure the fatigue behaviors from one or multiple nanocapacitors. Then, the time-of-flight secondary ion mass spectrometry (ToF-SIMS) is utilized to record the cation migration during the fatigue process. The conclusion was made that the degradation of ferroelectric properties was correlated with a local chemical change associated with the penetration of Cu⁺ ions from the electrode into the PZT structure, while the impact from oxygen vacancies was found to be insignificant. The work is well organized, and the conclusion is challenging the state-of-art points of view on the ferroelectric fatigue.

Thank you!

Although both AFM part and ToF-SIMS part are designed well, the logic behind the correlation of these two parts can be improved. In line 242-246 “As such, one can conclude, that fatigue in the studied system is mostly caused by effective doping of the PZT film due the migration of copper ions from the electrode.” The insignificance of one driving force, which is oxygen vacancies contribution in this work, will not directly make another driving force, which is cation migration, more significant.

Thank you for the excellent question! Indeed, our results indicate that Cu ions migration into the surface layer (up to 16 nm) of PZT is associated with the ferroelectric fatigue. They also indicate an absence of contributions from oxygen vacancies, as the oxygen concentration is undisturbed in the fatigued capacitors. However, we agree with the reviewer that this data is not sufficient on its own to claim that the Cu ion migration is the only cause of fatigue. Indeed, fatigue could also be caused by the nucleation and propagation of defect states or domain wall pinning in the Cu doped regions. However, it should be noted, that the explored cation migration is closely correlated with the onset of fatigue (confirmed by both AFM and ToF-SIMS studies). Therefore, it provides a new and unique perspective into the potential driving forces responsible for fatigue.

We have carefully edited the discussion and conclusion sections of the manuscript to reflect this point.

This also raises more questions:

1. Is Cu ion special in the ferroelectric fatigue processing? How about other cations used for electrode?

In this manuscript we studied fatigue with Cu-Au electrodes only. However, fatigue is known to be significantly dependent on the electrode material [see ref. 10 of the manuscript]. Nevertheless, our conclusions cannot be automatically extended to other materials. Though, it is a great topic for follow up research.

2. The DFT simulation in this work clearly shows that spontaneous polarization is significantly reduced with increasing Cu in $\text{Pb}_{1-x}\text{Cu}_x\text{TiO}_3$. However, Is it reasonable to compare the Cu ion concentration in PZT thin film with that in $\text{Pb}_{1-x}\text{Cu}_x\text{TiO}_3$?

We have now performed simulations in PZT instead of PTO. Updated simulations still show a decrease of spontaneous polarization and confirm the stability of the Cu doped PZT structure with a formation energy difference about -2.1 eV per Cu ion. Furthermore, the new calculations demonstrate a change in the

electronic structure of doped PZT from insulating to metallic at Cu concentrations exceeding 25%, which we believe is important for understanding the observed phenomena.

The simulation section of the manuscript has been updated to reflect this new data.

3. Suppose that Cu ion really is the origin of the fatigue, how does the limited Cu ion affect the whole PZT film or single nanocapacitor? According to the Fig.2b and line 226, the Cu ion penetration depth ranged from 8 nm to 16 nm and the PZT thin film is about 90 nm thick.

That is a great point! Indeed, ToF-SIMS measurements demonstrated ion migration occurred within the top 16 nm of a 90-nm-thick PZT film. Local hysteresis loops showed a 4-fold decrease of switchable piezoresponse signal and a 2-fold decrease of the coercive voltage. At first glance, this seems to be counterintuitive, as one expects changes of the functional properties comparable to the ratio of doped volume (~17%).

We rationalize this by suggesting that it is possible that the Cu-doped regions are associated with domain wall pinning, which has been directly observed in PZT capacitors in previous fatigue studies by PFM. We speculate that the inhomogeneity of doping, with the formation of small regions with lower spontaneous polarizations, could play a role as pinning centers. Those regions decelerate the motion of the domain walls, leading to the formation of frozen domain structure, which in turn decrease the switchable polarization. Alternatively, they can also support the formation of defect states with the same stoichiometry, which are undetectable for ToF-SIMS, but can significantly alter switchable polarization. A corresponding discussion has been added to the manuscript.

Reviewer #2 (Remarks to the Author):

Based on the combination of PFM and ToF-SIMS, the authors investigated polarization fatigue phenomenon of PZT capacitors in nanoscale. The authors observed the migration of Cu^+ ions into ferroelectric. The results are interesting.

Thank you!

However, I don't think the authors well demonstrated the correlation between Cu^+ ion migrations with polarization fatigue. The current version is not suitable for publication in Nat. Comm. Some of my comments are listed below.

1) though the authors observed the migration of Cu^+ ions in fatigued PZT nanocapacitors, there are no further evidences to prove that this migration results in ferroelectric fatigue. That is, this migration may be only one of experimental phenomena observed during fatigue process, but is not the reason of ferroelectric fatigue. Though the DFT simulation indicated that Cu doping in PbTiO_3 resulted in the decrease of polarization, I am not very sure whether the doping in simulation is the same or similar as the migration in experiment.

That is a good point. We agree that the provided data is not sufficient on its own to claim that the Cu ion migration is the only cause of fatigue. However, we note that the observed cation migration is a new phenomenon which is closely correlated with fatigue in the studied system. For instance, it can explain the formation of pinning centers or defect states, which in turn cause reductions of the switchable polarization. So, we believe it would be of interest to the scientific community. We have carefully edited the discussion section and conclusions of the manuscript to reflect this point.

2) In fact, the migrations of metal ions into insulating layer in MIM systems are well observed and studied, which is also regarded as one possible mechanism forming resistive devices (sometimes called conductive filaments). From this viewpoint, Cu^+ ion migration may result in the increase of conductivity of the PZT thin layer near the Cu electrode. This may result in the decrease of effective PZT thickness and thus the decrease of apparent coercive field.

Indeed, updated DFT simulations show that PZT doped by Cu with concentration exceeding 25% would change the film to metallic. In particular, this explains the decrease in the coercive field after cycling above 10,000 switching pulses.

We have added a corresponding discussion to the text.

3) about Figure 1, the authors should indicate which nanocapacitor was polarized in text.

This description has been added to the text.

4) about Figure 2a and relevant text, here 'PFM signals' means PFM phase signals? The hysteresis loops are usually obtained from PFM phase signals. Why the authors used the term 'piezoresponse' instead of 'PFM phase'? Why the change in PFM phase signals can respond the max. switchable response R_m ?

In the text and figure 2 we used piezoresponse (PR) signal $PR = A * \cos(\theta)$, where A is PFM amplitude and θ is PFM phase. During the switching PFM phase signal changes from $\pi/2$ to $-\pi/2$, which can be seen as abrupt change in PR on the loop. So in this case, values of the PR signal at extreme voltages can be used to measure material response and to further calculate the maximal switchable response R_m .

A corresponding clarification has been added to the text and figure 2 has been modified.

5) p7, line 129, 8V voltage is applied to 90nm thick PZT layer, how the authors get the field of 114 kV/mm?

Thank you for catching that. There was an error in the electric field calculation. It was fixed in the revised version of the manuscript in both the text and the figure. The right value of the field is 89 kV/mm.

6) Figure S3b, the hysteresis loops from samples 1 and 2 show very different coercive fields and a little different piezoresponse. Why the authors explained that 'The resulting hysteresis loops demonstrated behavior similar to the non-fatigued loops of the first capacitor (Supplemental Materials, Fig. S3)' in the main text (Line 154, p9)?

Indeed, the coercive field for capacitor #2 differs from the one measured on capacitor #1. However, the maximal switchable piezoresponse is similar, this demonstrates that the observed degradation in the PFM is due to the cycling of the capacitor, and not the degradation of the tip. We have modified the description in the text to reflect this.

7) about the equation in p9, Line 165. Since not all local area is covered by electrodes, so Scan/Scap is not equal to the number of nanocapictors. Thus the calculated Tscan is not correct.

At this point we need to disagree with the reviewer. T_{scan} was adjusted in a way that every individual capacitor (with average area S_{cap}) is switched by a certain number of cycles – N . In this approach, the average time the tip spends on an individual capacitor is $T_{cap} = T_{scan} * S_{cap}/S_{scan}$. Therefore, the number of switching cycles applied to a single cap will be an average $N = T_{cap} * f_{ac}$, where f_{ac} is bias frequency. That is how we derived equation (1) on page 9 of the manuscript.

Indeed, the AFM tip spends some time outside of the caps. However, what really matters is the ratio of the average capacitor area to the total scan area and not the area of all the scanned capacitors.

8) Figure 3a, p10, Line 182-183, 'However, increasing the number of cycles lead to the appearance of dark regions in the PFM amplitude signal, associated with a decrease in the piezoresponse of the cycled capacitors.' From Figure 3a, the local area after 10^6 cycles seems to have even more electro-active nanocapacitors than the local area after 10^4 cycles. Why?

To address this comment, we calculated the PFM amplitude signal averaged over the cycled region (Fig. R1). As one can see the piezoresponse actually decreases with increasing number of switching pulses. To show this, we have added the following figure to the supplemental materials.

Figure R1. Signal of PFM amplitude averaged over the regions cycled by different number of switching pulses.

8) in the whole text, what is the meaning of ‘...averaged over 5 nm surface layer of PZT’?

This is thickness of the averaging slab we used to represent the XY maps of the Cu⁺ and other species distribution right underneath the electrode.

9) Figure 5a, atom species should be labeled.

Figure 5a was updated and labels have been added.

10) in method part, the authors need to give more details on how to use AAO templates to form these nanoelectrodes.

The description has been extended.

REVIEWERS' COMMENTS:

Reviewer #1 (Remarks to the Author):

thank you for addressing my previous review comments. well done

Reviewer #2 (Remarks to the Author):

No more comments. The authors have well answered my questions.